# Stromal Cell-Derived Factor-1, P-Selectin, and Advanced Oxidation Protein Products with Mitochondrial Dysfunction Concurrently Impact Cerebral Vessels in Patients with Normoalbuminuric Diabetic Kidney Disease and Type 2 Diabetes Mellitus

**DOI:** 10.3390/ijms26104481

**Published:** 2025-05-08

**Authors:** Ligia Petrica, Florica Gadalean, Adrian Vlad, Danina Mirela Muntean, Daliborca Vlad, Victor Dumitrascu, Flaviu Bob, Oana Milas, Anca Suteanu-Simulescu, Mihaela Glavan, Sorin Ursoniu, Lavinia Balint-Marcu, Maria Mogos-Stefan, Silvia Ienciu, Octavian Marius Cretu, Roxana Popescu, Cristina Gluhovschi, Lavinia Iancu, Dragos Catalin Jianu

**Affiliations:** 1Division of Nephrology, Department of Internal Medicine II, “Victor Babes” University of Medicine and Pharmacy, No. 2, Eftimie Murgu Sq., 300041 Timisoara, Romania; petrica.ligia@umft.ro (L.P.); bob.flaviu@umft.ro (F.B.); milas.oana@umft.ro (O.M.); anca.simulescu@umft.ro (A.S.-S.); patruica.mihaela@umft.ro (M.G.); lavinia.balint@umft.ro (L.B.-M.); maria.mogos@umft.ro (M.M.-S.); silvia-ioana.ienciu@umft.ro (S.I.); gluh@umft.ro (C.G.); iuliana.alioani@umft.ro (L.I.); 2Centre for Molecular Research in Nephrology and Vascular Disease, Faculty of Medicine, “Victor Babes” University of Medicine and Pharmacy, No. 2, Eftimie Murgu Sq., 300041 Timisoara, Romania; vlad.adrian@umft.ro (A.V.); daninamuntean@umft.ro (D.M.M.); vlad.daliborca@umft.ro (D.V.); dumitrascu.victor@umft.ro (V.D.); sursoniu@umft.ro (S.U.); popescu.roxana@umft.ro (R.P.); jianu.dragos@umft.ro (D.C.J.); 3Centre for Cognitive Research in Neuropsychiatric Pathology (Neuropsy-Cog), Faculty of Medicine, “Victor Babes” University of Medicine and Pharmacy, No. 2, Eftimie Murgu Sq., 300041 Timisoara, Romania; 4Center for Translational Research and Systems Medicine, Faculty of Medicine, “Victor Babes” University of Medicine and Pharmacy, No. 2, Eftimie Murgu Sq., 300041 Timisoara, Romania; 5County Emergency Hospital Timisoara, 300723 Timisoara, Romania; 6Division of Diabetes, Nutrition, and Metabolic Diseases, Department of Internal Medicine II, “Victor Babes” University of Medicine and Pharmacy, No. 2, Eftimie Murgu Sq., 300041 Timisoara, Romania; 7Division of Pathophysiology, Department of Functional Sciences III, “Victor Babes” University of Medicine and Pharmacy, No. 2, Eftimie Murgu Sq., 300041 Timisoara, Romania; 8Division of Pharmacology, Department of Biochemistry and Pharmacology IV, “Victor Babes” University of Medicine and Pharmacy, No. 2, Eftimie Murgu Sq., 300041 Timisoara, Romania; 9Division of Public Health and History of Medicine, Department of Functional Sciences III, “Victor Babes” University of Medicine and Pharmacy, No. 2, Eftimie Murgu Sq., 300041 Timisoara, Romania; 10Division of Surgical Semiology I, Department of Surgery I, “Victor Babes” University of Medicine and Pharmacy, No. 2, Eftimie Murgu Sq., 300041 Timisoara, Romania; cretu.marius@umft.ro; 11Emergency Clinical Municipal Hospital Timisoara, 300041 Timisoara, Romania; 12Division of Cell and Molecular Biology II, Department of Microscopic Morphology II, “Victor Babes” University of Medicine and Pharmacy, No. 2, Eftimie Murgu Sq., 300041 Timisoara, Romania; 13Division of Neurology I, Department of Neurosciences VIII, “Victor Babes” University of Medicine and Pharmacy, No. 2, Eftimie Murgu Sq., 300041 Timisoara, Romania

**Keywords:** stromal cell-derived factor-1, P-selectin, advanced oxidation protein products, cerebral vessels, diabetic kidney disease

## Abstract

Diabetic kidney disease (DKD) displays a high prevalence of cardiovascular and cerebrovascular disease. Both the kidney and the brain share common pathogenic mechanisms, such as inflammation, endothelial dysfunction, oxidative stress, and mitochondrial dysfunction. The aim of this study was to establish a potential association of cerebral vessel remodeling and its related functional impairment with biomarkers of inflammation, oxidative stress, and mitochondrial dysfunction in the early stages of DKD in type 2 diabetes mellitus (DM) patients. A cohort of 184 patients and 39 healthy controls was assessed concerning serum and urinary stromal cell-derived factor-1 (SDF-1), P-selectin, advanced oxidation protein products (AOPPs), urinary synaptopodin, podocalyxin, kidney injury molecule-1 (KIM-1), and N-acetyl-β-(D)-glucosaminidase (NAG). The quantification of the mitochondrial DNA copy number (mtDNA-CN) and nuclear DNA (nDNA) in urine and peripheral blood was conducted using quantitative reverse transcription polymerase chain reaction (qRT-PCR). Using TaqMan tests, the beta-2 microglobulin nuclear gene (B2M) and the cytochrome b (CYTB) gene, which encodes subunit 2 of NADH dehydrogenase (ND2), were evaluated. The MtDNA-CN is the ratio of mitochondrial DNA to nuclear DNA copies, ascertained through the examination of the CYTB/B2M and ND2/B2M ratios. The intima-media thickness (IMT) measurements of the common carotid arteries (CCAs), along with the pulsatility index (PI) and resistivity index (RI) of the internal carotid arteries (ICAs) and middle cerebral arteries (MCAs), were obtained through cerebral Doppler ultrasonography (US). Additionally, the breath-holding index (BHI) was also measured by cerebral Doppler US. PI-ICAs, PI-MCAs, CCAs-IMT, RI-MCAs, and RI-ICAs demonstrated direct relationships with SDF-1, P-selectin, AOPPs, urine mtDNA, podocalyxin, synaptopodin, NAG, and KIM-1 while showing indirect correlations with serum mtDNA and the eGFR. In contrast, the BHI had negative correlations with SDF-1, P-selectin, AOPPs, urine mtDNA, synaptopodin, podocalyxin, KIM-1, and NAG while showing direct associations with serum mtDNA and the eGFR. In conclusion, a causative association exists among SDF-1, P-selectin, and AOPPs, as well as mitochondrial dysfunction, in early diabetic kidney disease (DKD) and significant cerebrovascular alterations in patients with type 2 diabetes mellitus and normoalbuminuric DKD, with no neurological symptoms.

## 1. Introduction

The burden imposed by diabetic kidney disease (DKD) on patients with diabetes mellitus (DM), either type 1 or type 2, respectively, translates into their referral to renal replacement therapies in a percentage of 30% and 40%, respectively [1].

Recently, apart from the classical phenotype of DKD, which associates albuminuria and renal function decline, a new phenotype has emerged, known as non-albuminuria diabetic kidney disease (NADKD). To date, this entity has an overall prevalence among patients with type 2 DM and renal function decline of about 45.6% and is defined by a urinary albumin-to-creatinine ratio (UACR) of <17–30 mg/g and an estimated glomerular filtration rate (eGFR) of <60 mL/min/m^2^ [2]. It is widely accepted that at least 10% of normoalbuminuric type 2 DM patients are affected by an early DKD, in whom the diagnosis of NADKD relies not only on the UACR but also on additional biomarkers [3]. A large body of evidence substantiates the intervention of proximal tubule (PT) in the mechanisms of albuminuria and reveals the fact that PT dysfunction is instrumental in albumin processing and elimination. These data have cornered the concept of diabetic tubulopathy in parallel with or most likely preceding albuminuria, a view that emphasizes PT intervention in the mechanisms of albuminuria [4,5].

NADKD displays a high prevalence of cardiovascular and cerebrovascular disease, despite the use of renin–angiotensin–aldosterone inhibitors and adequate metabolic control [6].

The kidney and the brain share similarities in terms of structure and function and are coordinated by common pathogenic mechanisms, such as inflammation, endothelial dysfunction, oxidative stress, and mitochondrial dysfunction [7]. The hemodynamic principles in both organs are strongly related to the macro- and microvascular modifications that consist of atherosclerosis, arteriosclerosis, and microangiopathy [8]. Moreover, it has been demonstrated that there is a dissociation in the time frame of endothelial dysfunction within the kidney and the brain, the latter being affected earlier in the course of DM [5,9].

The endothelial variability in the behavior of the two vascular territories is related to the intervention of several factors, such as advanced glycation end-products [10], angiogenic factors, platelet activation, oxidative stress, and mitochondrial dysfunction.

Stromal cell-derived factor-1 (SDF-1) is a member of the CXC chemokine family localized in the kidney in podocytes and tubules, contributing to inflammation and glomerulosclerosis, and subsequent albuminuria [11,12].

In addition, SDF-1 plays an important role in neurogenesis [13], has proinflammatory effects on the vascular endothelium [14], intervenes in the occurrence and progression of atherosclerosis, in neurogenesis, neuro-inflammation, and interacts with other neuromodulators [15], thus explaining its implication in the pathogenesis of cerebrovascular disease in patients with type 2 DM.

P-selectin, a member of the selectin family, mediates the activation of platelets on endothelial cells, thus being involved in the initiation of atherosclerosis [16,17]. Also, P-selectin is associated with the microvascular complications in the kidney, exerting its role from the early stages of DKD [18]. Cerebral small vessels are also affected by high levels of P-selectin, which induce inflammatory processes [19].

Advanced oxidation protein products (AOPPs) are derived from oxidation-modified albumin, the primary origin of AOPPs, but also from fibrinogen [20]. AOPPs are formed as the result of oxidative stress by the action of reactive oxygen species and by chloramines produced by myeloperoxidase in activated neutrophils [21]. AOPPs may induce the synthesis of inflammatory cytokines, including SDF-1α [21], and can be used as biomarkers of oxidative stress in the plasma and urine of type 2 DM patients [22].

Increased levels of plasma AOPPs were found as independently related to early stages of DKD [23], before the occurrence of albuminuria. Moreover, APPOs were associated with early subclinical atherosclerosis in both diabetic and non-diabetic chronic kidney disease [24,25].

It is worth underlining that the accumulation of AOPPs is significantly related to an inflammatory response in the kidney and vessels through the activation of NADPH oxidase. This observation points to the link between AOPPs and mitochondrial dysfunction and may explain their concurrent intervention in the pathogenesis of early DKD and associated cerebral vessel remodeling [23,26].

In type 2 DM, mitochondrial dysfunction implies complex modification in mitochondrial functions, especially the alterations of mitochondrial DNA (mtDNA) [27] evaluated in plasma [28,29,30] and urine [28,29,30].

In previous works, we demonstrated that mtDNA changes in serum and urine display a specific signature in relation to inflammation within the diabetic kidney [29] and the cerebral vessels in neurologically asymptomatic type 2 DM patients with normoalbuminuric DKD [30]. Mitochondrial dysfunction evaluated by mtDNA damage intervenes in atherosclerosis and cerebral microvascular complication, thus increasing the risk for stroke and its severe consequences in terms of motor impairment and cognitive disabilities [31,32].

The aim of our study was to evaluate the interrelation of inflammation, oxidative stress, endothelial dysfunction, and mitochondrial dysfunction with proximal tubule dysfunction and podocyte damage in normoalbuminuric DKD. Also, this study was conducted with the view to establish a potential association of cerebral vessel remodeling and its related functional impairment with biomarkers of inflammation, oxidative stress, and mitochondrial dysfunction in the early stages of DKD in type 2 DM patients.

## 2. Results

### 2.1. Clinical and Biological Data of Patients with Type 2 DM and Controls

The clinical and biological date of patients with type 2 DM and controls are presented in Table 1 as medians and IQR. In Table 1, *p* values were corrected for multiple comparisons. The data provided reveal significant difference between the serum and urine biomarkers of PT dysfunction (KIM-1, NAG), podocyte damage (synaptopodin, podocalyxin), mitochondrial dysfunction (mtDNA), inflammation (SDF-1), endothelial dysfunction (P-selectin), and oxidative stress (AOPPs).

The biomarkers studied were increased in serum and urine in all groups of patients vs. controls. Also, in Table 1 are presented the differences concerning the IMT-CCAs, PI-ICAs, RI-ICAs, PI-MCAs, and RI-MCAs, which were increased in patients with type 2 DM vs. controls. By contrast, the BHI was decreased in patients as compared to healthy control subjects, a fact that reveals a decreased vasodilatory capacity of the cerebral vessels in diabetic patients in response to a vasodilatory stimulus, such as hypercapnia.

### 2.2. Variations in Blood and Urine of SDF-1, P-Selectin, and AOPPs Are Associated with mtDNA Changes in Early DKD

Univariable regression analysis showed that serum SDF-1 correlated negatively with the eGFR and serum mtDNA (*p* < 0.001) and directly with UACR, KIM-1, NAG (*p* < 0.001), synaptopodin, and podocalyxin (*p* < 0.001). Urinary SDF-1 correlated directly with urinary mtDNA, UACR, KIM-1, NAG (*p* < 0.001), synaptopodin, podocalyxin (*p* < 0.001), and indirectly with the eGFR (*p* < 0.001).

Serum P-selectin correlated indirectly with serum mtDNA and the eGFR (*p* < 0.001) and directly with UACR, KIM-1, NAG (*p* < 0.001), synaptopodin, and podocalyxin (*p* < 0.001). Urinary P-selectin correlated directly with urinary mtDNA (*p* < 0.001), UACR, KIM-1, NAG (*p* < 0.001), synaptopodin, and podocalyxin (*p* < 0.001), while there was a negative correlation with the eGFR (*p* < 0.001).

AOPPs in blood and urine were significantly increased in patients with type 2 DM, displaying an ascending slope from normo-to-micro and macroalbuminuria, respectively. This trend parallelled the decreased levels of serum mtDNA and the eGFR (*p* < 0.001) and the increased levels of UACR, KIM-1, NAG (*p* < 0.001), synaptopodin, podocalyxin (*p* < 0.001), and urinary mtDNA (*p* < 0.001) (Table 2). The correlations between serum mtDNA and urinary mtDNA and between serum and urinary SDF-1, P-selectin, and AOPPs, respectively, are shown in Figure 1a–f.

In multivariable regression analysis, the models provided results that revealed that serum mtDNA correlated directly with the eGFR and negatively with serum P-selectin and serum SDF-1 (R^2^ = 0.6564; *p* < 0.0001). Urinary mtDNA correlated directly with podocalyxin, urinary P-selectin, and urinary SDF-1 (R^2^ = 0.6280; *p* < 0.0001) (Table 3).

### 2.3. Cerebrovascular Hemodynamic Indices Are Associated with Changes in Blood and Urine Levels of SDF-1, P-Selectin, AOPPs, mtDNA, and Renal Tubular and Glomerular Biomarkers

In univariable regression analysis, IMT-CCAs, PI-ICAs, RI-ICAs, PI-MCAs, and RI-MCAs correlated negatively with the eGFR and serum mtDNA and directly with serum and urinary SDF-1, P-selectin, AOPPs, UACR, KIM-1, NAG, and urinary mtDNA (Table 4).

The models yielded by multivariable regression analysis for the cerebral hemodynamic indices showed that the IMT-CCAs correlated negatively with the eGFR and directly with the UACR, synaptopodin, serum P-selectin, and serum AOPPs (*p* < 0.0001; R^2^ = 0.7020). The PI-ICAs correlated indirectly with the eGFR and directly with serum P-selectin and serum SDF-1 (*p* < 0.0001; R^2^ = 0.5113). The PI-MCAs had a negative correlation with the eGFR and direct correlations with serum P-selectin and serum SDF-1 (*p* < 0.0001; R^2^ = 0.5886).

The RI-ICAs were included in a complex model that displayed an indirect correlation with the eGFR and direct correlations with NAG, serum P-selectin, serum SDF-1, and UACR (*p* < 0.0001; R^2^ = 0.7629). The RI-MCAs were included in a model that showed a negative correlation with the eGFR and direct correlations with NAG, KIM-1, synaptopodin, podocalyxin, serum P-selectin, and serum SDF-1 (*p* < 0.0001; R^2^ = 0.8482). The CVR assessed by BHI revealed that the BHI correlated directly with eGFR and negatively with podocalyxin, serum P-selectin, and serum SDF-1 (*p* < 0.0001; R^2^ = 0.7897) (Table 5).

## 3. Discussion

According to the data in the literature, our study is the first study that aims to establish a potential concurrent association between biomarkers of inflammation, mitochondrial dysfunction, and oxidative stress and early renal and cerebrovascular involvement in normoalbuminuric DKD in patients with type 2 DM. This study provides data showing that SDF-1, P-selectin, and AOPPs may intervene in early DKD along with changes in the blood and urine of mtDNA, as well as with the biomarkers of PT dysfunction and podocyte damage. Furthermore, this study reveals that cerebral vessels undergo atherosclerotic, arteriosclerotic, and microvascular remodeling in relation to the biomarkers studied. It should be underlined that SDF-1, P-selectin, and AOPPs imposed a significant impact upon the functionality of the cerebral vessels as demonstrated by the impaired cerebrovascular reactivity.

### 3.1. SDF-1, P-Selectin, and AOPP Variations in Blood and Urine in Conjunction with mtDNA Are Interrelated with Early DKD in Patients with Type DM

*Stromal cell derived factor-1* exerts an essential regulatory role in the initiation and progression of DKD [33]. In a study conducted by Lu et al., serum SDF-1 levels were higher in patients with type 2 DM and early DKD, as assessed by cystatin C levels. The authors showed that SDF-1 was positively related to cystatin C and was a significant independent contributor to DKD [34]. Moreover, their study underlined that tubular injury may precede the occurrence of glomerular injury due to the fact that cystatin C is a reliable indicator of renal tubular damage [34], an observation that holds true in other studies too [29,35].

In line with the study conducted by Lu et al., we found that SDF-1 levels correlated directly with the UACR and negatively with the eGFR assessed by the combined formula that uses serum creatinine and cystatin C [34]. In addition, in our study, SDF-1 correlated significantly with mtDNA, biomarkers of PT dysfunction, and podocyte injury, even in normoalbuminuric patients with type 2 DM.

Despite controversial data suggesting that SDF-1 may exert renoprotective effects in DKD [36], the vast majority of data underlines the negative impact of SDF-1 on DKD progression [33]. It has been shown that SDF-1 produced mainly by the podocytes enhanced proteinuria and glomerulosclerosis in a mouse model of type 2 DM. The authors show that the transient blockade of SDF-1 prevents the progression of proteinuria and glomerulosclerosis in the early stages of DKD [12,37]. The results of our study are in line with these observations supported by the direct correlations of SDF-1 with PT dysfunction and podocyte damage biomarkers. Also, SDF-1 correlated negatively with the eGFR, an observation that is similar to the results of the study conducted by Huang et al. [38]. The data deriving from the references consulted are lacking information concerning the relation of SDF-1 with mtDNA, and, thus, we could not compare our results with other studies. In our study, SDF-1 correlated with serum and urine mtDNA and intervened concurrently in the podocyte and PT dysfunction in normoalbuminuric DKD.

*P-selectin*, as a platelet-derived proinflammatory factor, is elevated in patients with type 2 DM and is positively correlated with the severity of DKD [39]. In the study conducted by Al-Rubeaan et al., P-selectin was significantly associated with a decline in the eGFR in diabetic patients with type 2 DM and normal albuminuria excretion. The authors link this observation with the role of P-selectin in the adhesion of leucocytes to the endothelium, thus leading to endothelial dysfunction and subsequent impairment in microangiopathic changes and decline in the eGFR [40]. Also, the authors conclude that elevated levels of P-selectin in normoalbuminuric patients may serve as a biomarker able to predict renal injury before albuminuria [40]. Similar data were provided by the study carried out in type 2 DM patients by Siddqui et al., who reported that P-selectin may be instrumental in the development of microvascular complications within the kidney [18].

In our study, P-selectin paralleled the trend of SDF-1 and correlated negatively with the eGFR and serum mtDNA and directly with urinary mtDNA and the biomarkers of PT dysfunction and podocyte injury, independently of the levels of albuminuria.

*Advanced oxidation protein products* are found as the result of oxidative stress by the action of reactive oxygen species and by chloramines (produced by myeloperoxidase in activated neutrophils) on serum proteins [22].

In our study, plasma and urinary AOPPs were increased and correlated negatively with the eGFR and serum mtDNA and directly with the biomarkers of PT dysfunction and podocyte injury even in normoalbuminuric patients with type 2 DM. These observations may be explained according to the data provided by Iwao et al., who showed that the variations in the levels of plasma AOPPs may be related to a certain extent by an imbalance between the increased formation of AOPPs and decreased urinary eliminations [41].

In our study, AOPPs correlated with mtDNA variations, thus showing a strong link between mitochondrial dysfunction and oxidative stress. Similar data were reported in experimental studies in which the accumulation of AOPPs may promote an inflammatory response in the diabetic kidney, most likely due to the activation of NADPH oxidase [26] and through AOPP-induced mitochondrial dysfunction and oxidative stress [42,43]. Also, the association of increased levels of AOPPs with P-selectin and SDF-1 strengthens the relation between AOPPs and podocytes [44], endothelial cells within the kidney, and other similar vascular territories [42,43]. Moreover, Liang et al. reveal that increased plasma levels of AOPPs were an independent risk factor for endothelial dysfunction in type 2 DM patients with non-albuminuria [23]. The results of our study show that both plasma and urinary AOPPs were correlated with inflammation and endothelial dysfunction across all groups of patients studied, irrespective of the level of albuminuria and of renal function decline.

### 3.2. SDF-1, P-Selectin, and AOPPs Are Major Players in the Mechanisms of Cerebrovascular Remodeling in Patients with Type 2 DM and Normoalbuminuric DKD

In previous studies conducted by us with regard to the brain–kidney axis, we demonstrated that cerebrovascular remodeling precedes the occurrence of renal involvement in normoalbuminuric DKD, expressed by early PT dysfunction and podocyte damage. These later phenomena were extensively studied by us, and we showed that endothelial dysfunction in the cerebral vessels precedes endothelial dysfunction in the kidney in normoalbuminuric type 2 DM patients [5]. Therefore, we hypothesized that there are distinct endothelial territories within the kidney and the brain [10].

Cerebral vessels undergo severe modifications in terms of atherosclerosis and arteriosclerosis, far more severe even in normoalbuminuric type 2 DM patients [6] and may remain silent, with no clinical symptoms in the long term, which was the case with the patients included in our studies [9,30].

Cerebral vessel remodeling, including subclinical lesions, may be assessed by neurosonologic methods, which are supported by several parameters, such as carotid intima-media thickness (IMT) [45,46], a reliable indicator of subclinical atherosclerosis, even in normoalbuminuric type 2 DM patients [9,47].

*Stromal cell derived factor-1* is a component of the immune system within the nervous system, exerting its role of chemokine upon glial and endothelial cells [15]. Most importantly, the SDF-1 beta isoform is selectively expressed by endothelial cells of cerebral microvessels, thus explaining its intervention in microvascular structural and functional modifications [48]. In addition, SDF-1 is expressed in human atherosclerotic plaques, data provided by the studies conducted by Al-Younes et al. and by Merckelbach et al., who showed that SDF-1 could be involved in the pathology of atherosclerosis [49,50]. Also, the same studies reveal that SDF-1 can activate platelets and may play a role in the formation of platelet-rich thrombus and plaque disruption [49].

In our study, atherosclerotic and arteriosclerotic changes in the cerebral vessels, as demonstrated by the increased levels of the neurosonologic parameters IMT-CCAs, PI-ICAs, RI-ICAs, PI-MCAs, and RI-MCAs, correlated directly with SDF-1, an association that might reinforce the intervention of this chemokine in cerebrovascular remodeling. Moreover, in our study, the cerebrovascular reactivity, strongly related to the capacity of cerebral microcirculation to respond through vasodilatation to a vasodilating stimulus such as hypercapnia, was significantly impaired in patients with type 2 DM and early DKD. Under these circumstances, the BHI, as an accurate parameter that indicates the cerebrovascular reactivity, correlated indirectly with SDF-1, an observation that emphasizes the hypothesis according to which this chemokine may exert an intimate role within the cerebral microvascular territory. Of interest, SDF-1 exerts an important function in modulating mitochondrial respiration through regulating mitochondrial oxidation phosphorylation, ATP production, and mitochondrial content [51], thus intervening in platelet activation and thrombus formation [52].

*P-selectin*, a member of the adhesion molecular family, is a marker of platelet activation and of endothelial dysfunction, with an active intervention in the atherosclerotic and arteriosclerotic vascular processes [17]. Ultrasonographic assessments of the cerebral vessels revealed a significant association of P-selectin with an atherosclerotic wall thickness of the carotid arteries measured by increased levels of carotid IMT and arterial wall stiffness [16]. In our study, IMT-CCAs, as well as the PIs and RIs in the ICAs and MCAs, respectively, correlated directly with P-selectin, a fact that is aligned with the above-mentioned studies.

Similarly to the microvascular changes within the kidney [18], cerebral small vessels are associated with variations in the levels of P-selectin. Cerebral small vessel disease has an underlying inflammatory process related to endothelial dysfunction mediated by proinflammatory cytokines, including P-selectin [19]. In our patients with type 2 DM, the CVR was impaired in NADKD, a phenomenon expressed by the decreased BHI and by the correlation of the later parameter with the levels of P-selectin.

*Advanced oxidation protein products* are interrelated with other chemokines, such as SDF-1 α and monocyte chemoattractant protein-1 and intervene in vascular inflammatory processes [53]. Carotid intima-media thickness and carotid atherosclerosis are closely influenced by AOPPs [20,54], especially in patients with chronic kidney disease [24,55].

In our study, IMT-CCAs correlated with the levels of AOPPs, a phenomenon that also holds true for the PIs and RIs in the ICAs and the MCAs, modifications that align with the results of the study conducted by Bagyura et al., conducted in an asymptomatic population, which shows that AOPPs are a marker of oxidative stress by impacting the carotid IMT [25]. Similar studies have provided evidence concerning the relation between AOPPs and other markers of oxidative stress and carotid IMT [56]. It has been postulated that AOPPs are an independent risk factor for endothelial dysfunction and a strong predictor for impaired endothelial-dependent vasodilatation [23]. Our results concerning the CVR and the decreased BHI, although limited by not measuring blood CO2 during the maneuver, show an indirect correlation with the increased levels of AOPPs, an observation that supports the AOPP-induced endothelial dysfunction and impaired vasodilating capacity of the cerebral vessels. In the study conducted by Liang et al., increased plasma AOPPs were an independent risk factor for endothelial dysfunction and considered an early marker of vasculopathy in normoalbuminuric patients with type 2 DM [23]. The authors demonstrated that plasma concentrations of AOPPs were closely correlated with brachial arterial flow-mediated dilation. It appears that the vasculopathy develops early in the setting of type 2 DM, in the stage of normoalbuminuria [5,10,57].

In contrast with SDF-1 and P-selectin, in which the literature lacks information with regard to their relation to mitochondrial dysfunction, it has been reported that AOPPs are important mediators of oxidative stress and can stimulate the NADPH oxidase respiratory burst, thus establishing a connection with impaired mitochondrial function [58]. In our study, the variations in the levels of mtDNA, as a token of mitochondrial dysfunction, correlated with AOPPs in patients with type 2 DM and important cerebral vessel modifications.

Our study has several limitations. First, this is a cross-sectional exploratory study that only allows for the conclusions of associations between SDF-1, P-selectin, AOPPs and the variables studied, but not of causality. Second, blood CO2 levels were not measured during the maneuver of breath holding, thus decreasing the accuracy of the method. Third, the medication that the patients were under, such as ACEIs/ARBs, statins, and SGLT2 inhibitors/GLP-1 agonists, might have precluded an accurate interpretation of data.

The strengths of our study reside in several aspects as this attempts to evaluate a concurrent intervention of inflammation, oxidative stress, and mitochondrial dysfunction in the kidney and cerebrovascular remodeling in patients with type 2 DM and normoalbuminuric DKD. This is a human study conducted on a cohort of patients with type 2 DM in whom we demonstrated that SDF-1, P-selectin, and AOPPs may be considered accurate indicators of early renal and cerebrovascular involvement in the course of type 2 DM. The significant correlations of these biomarkers with biomarkers of PT dysfunction, podocyte damage, and mitochondrial dysfunction corner a particular inflammatory profile in patients with normoalbuminuric DKD. Also, SDF-1, P-selectin, and AOPPs, associated with the structural and functional modification of cerebral vessels in type 2 DM patients, support their diagnostic value in characterizing intimate pathways in the occurrence of cerebrovascular remodeling in the early stages of DKD.

## 4. Materials and Methods

### 4.1. Cohort/Inclusion/Exclusion Criteria

This study encompassed a cohort of 184 patients (93 males, 91 females) diagnosed with type 2 diabetes mellitus (DM), selected from a total of 278 consecutive patients visiting the Outpatient Departments of Nephrology and Diabetes and Metabolic Diseases from January 2023 to December 2024. The sample size calculation was not performed as this was an exploratory study. The subjects, aged 50 to 78 years, were evaluated through personal visits and review of their records. The inclusion criteria were duration of diabetes mellitus exceeding 5 years and to be receiving treatment with either oral antidiabetic agents (including metformin, gliclazide, SGLT2 inhibitors, or GLP-1 receptor agonists), insulin, angiotensin-converting enzyme inhibitors, angiotensin receptor blockers, or statins. Hypertensive individuals were represented by 66.84% of the patients included in this study. Individuals exhibiting inadequate glycemic regulation (characterized by a HbA1c level exceeding 10%) or possessing a history or manifestations of cerebrovascular illness or coronary artery disease were excluded from the trial. A total of 184 individuals with type 2 diabetes mellitus were classified into three groups: 64 patients with normoalbuminuria (UACR < 30 mg/g), 59 patients with microalbuminuria (UACR 30–300 mg/g), and 61 patients with macroalbuminuria (UACR > 300 mg/g). A control group comprising 39 age- and gender-matched healthy adults was included, all of whom had no history of renal illness and were screened to exclude diabetes or pre-diabetes, as indicated by a HbA1c level of 5.6% or below, according to general practitioner records.

### 4.2. Laboratory Assessments

Serum and urine samples from both patients and controls were stored at −80 °C and thawed before analysis. Urinary biomarkers were assessed using the first morning urine sample and expressed relative to the urinary creatinine ratio. The biomarkers examined were evaluated utilizing the ELISA technique, as detailed below:-Human SDF-1 (stromal cell-derived factor-1) (Catalog No. E-EL-H0052, Elabscience, Houston, TX, USA); sensitivity: 0.1 ng/mL; detection range: 0.16–10 ng/mL; and coefficient of variation (CV) < 10%.-Human SELP (P-Selectin) (Catalog No. E-EL-H0917, Elabscience, Houston, TX, USA); sensitivity: 0.1 ng/mL; detection range: 0.16–10 ng/mL; and CV < 10%.-OxiSelectTM AOPP Assay Kit (Catalog Number STA-318, Cell Biolabs, San Diego, CA 92126, USA); AOPP-HSA Positive Control [Part No. number_2]: One 100 μL tube containing 7.5 mg/mL of AOPP-Human Serum Albumin with 0.14 μmol AOPP/mg proteins.-Podocyte injury biomarkers included synaptopodin (Catalog No. abx055120, Abbexa, Cambridge, UK), with a sensitivity of 0.10 ng/mL, a detection range of 0.156–10 ng/mL, and a coefficient of variation (CV) of less than 10%. Podocalyxin (Catalog No. E-EL-H2360, Elabscience, Houston, TX, USA) has a sensitivity of 0.1 ng/mL, a detection range of 0.16–10 ng/mL, and a coefficient of variation (CV) of less than 10%.

Proximal tubule (PT) dysfunction biomarkers included kidney injury molecule-1 (KIM-1, Catalog No. E-EL-H6029, Elabscience, Houston, TX, USA; sensitivity: 4.69 pg/mL; detection range: 7.81–500 pg/mL; and CV < 10%) and N-acetyl-β-(D)-glucosaminidase (NAG, Catalog No. E-EL-H0898, Elabscience, Houston, TX, USA; sensitivity: 0.94 ng/mL; detection range: 1.56–100 ng/mL; and CV < 10%).

Serum and urine samples were analyzed in triplicate, adhering to the manufacturer’s guidelines. The eGFR was calculated using the combined serum creatinine–cystatin C (CKD-EPI equation), following the KDIGO 2024 Guidelines for the Evaluation and Management of CKD [59].

### 4.3. Evaluation of mtDNA

qRT-PCR (CFX Connect-Biorad Laboratories, Carlsbad, CA, USA) was utilized for the evaluation of mtDNA-CN and nuclear DNA (nDNA) in peripheral blood and urine. TaqMan assays assessed the cytochrome b (CYTB) gene, subunit 2 of NADH dehydrogenase (ND2), and beta 2 microglobulin nuclear gene (B2M). Primers for the CYTB and ND2 genes were utilized as target sequences for the assessment of mtDNA. B2M was utilized as an internal reference gene for nDNA analysis. MtDNA-CN was defined as the ratio of the number of mtDNA/nDNA copies through analysis of the CYTB/B2M and ND2/B2M ratios. Genomic DNA was obtained from biological samples using PureLink™ Genomic DNA Kit (Life Technologies, Carlsbad, CA, USA), following the protocol from the manufacturer’s brochure. The concentration of extracted DNA was measured by fluorimetric quantification (Qubit, Invitrogen, Thermo Fisher Scientific, Waltham, MA, USA). For real-time quantitative tests, the DNA samples were diluted to 10 ng/μL. Real-time quantitative polymerase chain reaction was applied using TaqMan Universal PCR master mix and TaqMan primers (Applied Biosystems, Thermo Fisher Scientific, Waltham, MA, USA). Samples were run in triplicate, in MicroAmpR Optical 96-well reaction plates, each well containing 9 μL of diluted DNA, 1 μL primers, and 10 μL of master mix. The total reaction volume was 20 μL. The thermal cycle profile was 2 min at 500 °C for UNG incubation, 10 min at 950 °C for polymerase activation, 40 cycles of 15 s at 950 °C (denaturation), and 1 min at 600 °C (annealing and extension).

For each run, there was a melting curve analysis to check nonspecific products. Relative mtDNA quantification was carried out as previously described [60]. The results were analyzed using the comparative Ct method. ΔCt (values of Δ cycle thresholds) in the sample were calculated by subtracting the values for the reference gene from the sample Ct and normalizing to nuclear DNA. 2-ΔCt was obtained, and the results were expressed as relative quantification. The obtained values were normalized to nuclear DNA and are reported as the number of copies per nuclear DNA (mtDNA/nDNA). Urinary values were normalized to urine creatinine.

### 4.4. Neurosonologic Ultrasound Methods

Due to the remodeling of cerebral blood vessels, increased resistance in the examined vessels can be evaluated using several hemodynamic indices, including the pulsatility index (PI) and the resistance index (RI). Low-resistance vessels, like the ICAs, and the MCAs exhibit elevated PIs and RIs. These increased levels of both indices indicate heightened vascular resistance and diminished vasodilation capacity, which are functional manifestations of conditions such as cerebral atherosclerosis, arteriosclerosis, and microangiopathy [10,46,61]. Cerebrovascular reactivity (CVR) refers to the increase in blood flow velocity in response to a vasodilatory stimulus [62]. In patients with DM-T2, CVR is often diminished due to existing vasodilation. As a result, this impairment reduces the body’s ability to respond effectively to changes in cerebral blood flow through its autoregulatory mechanisms [10,63]. The cerebrovascular hemodynamic indices were assessed by a neurologist who was unaware of the clinical and biological data of both subject groups. The evaluation utilized a high-resolution ultrasound machine (Esaote MyLab 8, Genoa, Italy) equipped with a Color Ultrasound System, featuring two transducers: one with a selectable frequency range of 1.7 to 4 MHz (multifrequency sectorial transducer-phased array) and another with a frequency range of 3.6 to 16 MHz (linear transducer). The cerebrovascular ultrasound technique was applied as previously described [30]. We will briefly outline the methods utilized.

#### 4.4.1. Carotid Artery Intima-Media Thickness (IMT)

The carotid artery IMT was evaluated bilaterally in the common carotid arteries (CCAs). IMT refers to the distance between the luminal–intimal interface and the media-adventitial interface of the carotid arteries. This measurement is displayed in a double-line pattern using ultrasound in brightness mode (B-mode) during a longitudinal view of the carotid artery [45]. Each patient and control subject underwent three IMT measurements, and the median value from these measurements was used for further analysis. The standard cut-off point for IMT was established at less than 1.0 mm.

#### 4.4.2. Pulsatility Index (PI) and Resistivity Index (RI)

The PI and the RI were evaluated bilaterally in the ICAs using continuous wave Doppler ultrasound at a frequency of 4 MHz. Simultaneously, the MCAs were assessed with pulsed wave Doppler ultrasound at a frequency of 2 MHz. These hemodynamic indices were calculated automatically using specific formulas, which include Gosling’s PI, formulated as (systolic flow velocity–diastolic flow velocity)/mean flow velocity (with a standard value of <1), and Pourcelot’s RI, calculated as (systolic flow velocity–diastolic flow velocity)/systolic flow velocity (with a standard value of <0.7) [46].

#### 4.4.3. Cerebrovascular Reactivity

The CVR—which measures the responsiveness of cerebral blood vessels to vasodilation stimuli—was evaluated using the transcranial Doppler breath-holding test (BHT) in the MCAs bilaterally. The procedure began after the participants breathed room air for approximately four minutes, followed by a breath-holding phase lasting 30 s at the end of a normal expiration. Hemodynamic parameters, including mean flow velocity (MFV), systolic flow velocity, and diastolic flow velocity, were monitored at rest, during the breath-holding maneuver, and, at the end of the BHT, at the peak of hypercapnia, which served as the vasodilatory stimulus. To allow the MFV to return to baseline levels, the maneuver was repeated after a 2–3 min rest period. The mean MFV readings in the MCAs and the mean breath-holding index (BHI) were then calculated. The BHI is computed as the percentage increase in MFV in the MCAs during the breath-holding maneuver, divided by the duration of breath holding in seconds. It is expressed with the formula [(Vbh − Vr/Vr) × 100 s^−1^], where Vbh is the MFV in the MCAs at the end of breath holding, Vr is the MFV at rest, and s^−1^ indicates per second of breath holding. The standard value for the BHI is 1.2 ± 0.6 [63].

### 4.5. Statistical Analysis

The statistical methods were applied in accordance with the requirements of case-series exploratory studies. Clinical and biological data are presented as medians and interquartile ranges (IQRs) for variables that exhibit a skewed distribution. Differences between subgroups were analyzed using the Mann–Whitney U test for comparisons between two groups and the Kruskal–Wallis test for comparisons among four groups based on the distribution of values. Regression analyses were performed to assess the significance of the relationships between cerebrovascular hemodynamic indices and serum and urinary levels of SDF-1, P-selectin, AOPPs, mtDNA, as well as other continuous variables, including synaptopodin, podocalyxin, KIM-1, NAG, UACR, and the eGFR. A univariable regression analysis was conducted to evaluate the significance of the relationships among continuous variables across all four groups, which included pooled data from normal, microalbuminuric, and macroalbuminuric patients and healthy controls. Only those variables that showed significance in the univariable regression analysis were included in the subsequent multivariable regression analysis models. We used variance inflation factor (the **vif** command) after the regression to check for multicollinearity. Usually, a variable with a VIF value greater than 10 may warrant further research. The threshold for statistical significance was set at *p* < 0.05, and this study was conducted using Stata 18 (StataCorp, College Station, TX, USA).

## 5. Conclusions

This study demonstrates a concurrent association of SDF-1, P-selectin, and AOPPs, in conjunction with mitochondrial dysfunction, in the initiation of renal tubular and podocyte lesions in early DKD. Furthermore, the association of these biomarkers with neurosonologic parameters reflects incipient cerebrovascular modifications in patients with type 2 DM and nomoalbuminuric DKD, with no neurological symptoms. SDF-1, P-selectin, and AOPPs, along with mtDNA, could be used as a panel of biomarkers that may predict subclinical renal and cerebrovascular involvement in normoalbuminuric patients with type 2 DM.

## Figures and Tables

**Figure 1 ijms-26-04481-f001:**
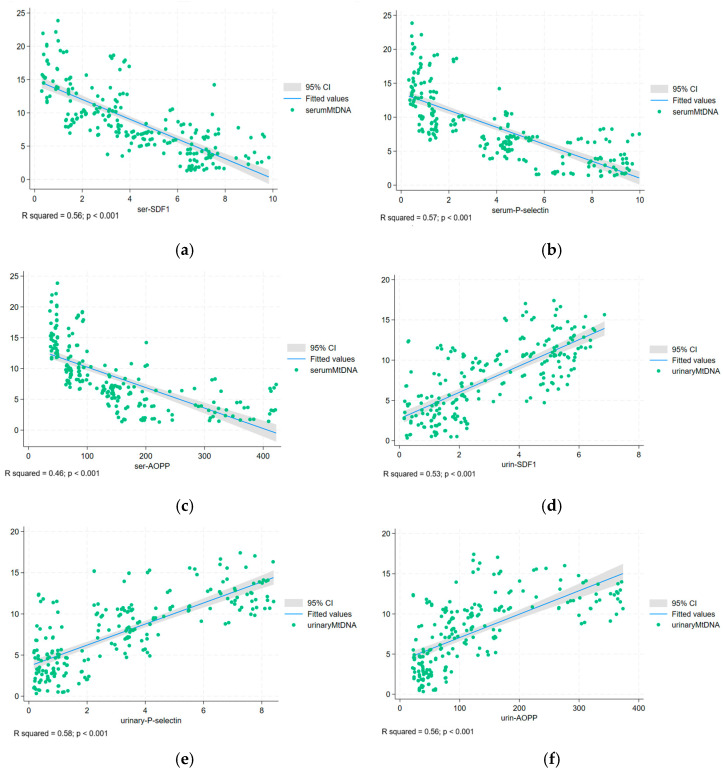
The correlations between (**a**) serum mtDNA and serum SDF-1; (**b**) serum mtDNA and serum P-selectin; (**c**) serum mtDNA and serum AOPP; (**d**) urinary mtDNA and urinary SDF-1; (**e**) urinary mtDNA and P-selectin; (**f**) urinary mtDNA and urinary AOPP.

**Table 1 ijms-26-04481-t001:** Clinical and biological data of patients with type 2 DM and controls.

Parameter	Controls (N = 39)	Normoalbuminuric Patients (N = 64)	Microalbuminuric Patients (N = 59)	Macroalbuminuric Patients (N = 61)	*p*-Value
**N**	39 (17.5%)	64 (28.7%)	59 (26.5%)	61 (27.4%)	<0.0001
**Age (years)**	67.47 (64; 69)	68.33 (65; 72)	69.23 (65; 74)	69.8 (67; 73)	<0.0001
**BMI (kg/m^2^)**	25.2 (23; 27) ^#,^*	29.26 (26.5; 31.5) ^⌂^	31.47 (28; 34)	30.8 (27; 32)	<0.0001
**SBP (mmHg)**	117.17 (110; 120) ^#,^*	138.37 (120; 150)	141.42 (130; 152.5)	147.8 (140; 165)	<0.0001
**DBP (mmHg)**	69 (65; 70) ^#,^*	79.15 (70; 90)	80.2 (70; 90)	81.5 (70; 90)	<0.0001
**DM duration (years)**	-	15.22 (10; 16.5)	17.73 (12; 23) ♦	21.04 (16; 26)	<0.0001
**Serum creatinine (mg/dL)**	0.80 (0.71–0.85) ^#,^*	1.17 (1.11–1.25)	1.20 (1.08–1.28) ^■^	1.26 (1.15–1.50)	<0.0001
**eGFR (mL/min/1.73 m^2^)**	83.33 (80.96–87.85) ^#,^*	57.13 (56.17–58.51) ^♣^	54.66 (52.14–58.51) ^⁑^	50.60 (44.59–56.73)	<0.0001
**HbA1c (%)**	5.00 (4.80–5.20) ^#,^*	7.00 (6.55–7.90) ^♣^	8.20 (7.50–9.60)	8.20 (7.90–9.10)	<0.0001
**Cholesterol (mg/dL)**	122 (115–153) ^#^	160(134–193)	167(132–202) ^⁑^	201(172–223)	<0.0001
**Triglycerides (mg/dL)**	97(88–102) ^#,^*	145(108–172)	150(119–215) ^♦^	194(156–281)	<0.0001
**UACR (mg/g)**	11(10–17) ^#,^*	23(15–27) ^♣^	89(61–131) ^■^	693(463–1312)	<0.0001
**Synaptopodin/cr (mg/g)**	0.84 (0.62–0.95) ^#,^*	1.73 (1.28–1.90) ^♣^	2.41 (2.20–2.57) ^■^	5.36 (3.08–8.25)	<0.0001
**Podocalyxin/creat (mg/g)**	0.26 (0.22–0.50) ^#,^*	1.15 (1.03–1.25) ^♣^	3.40 (3.25–4.19) ^■^	8.20 (7.02–9.07)	<0.0001
**NAG/creat (ng/g)**	2.52 (2.26–2.86) ^#,^*	4.92 (3.04–9.32) ^♣^	11.68 (10.51–17.77) ^■^	17.78 (17.18–19.79)	<0.0001
**KIM-1/creat (pg/g)**	43.82 (37.83–47.36) ^#,^*	78.77 (66.93–94.42) ^♣^	141.33 (130.34–149.14) ^■^	425.75 (338.06–482.49)	<0.0001
**Serum-P-selectin**	0.52 (0.46–0.71) ^#,^*	1.33 (1.17–1.49) ^♣^	4.45 (4.16–4.64) ^■^	8.37 (7.20–9.20)	<0.0001
**Urinary-P-selectin**	0.33 (0.24–0.41) ^#,^*	1.00 (0.80–1.31) ^♣^	3.39 (3.09–3.64) ^■^	7.07 (5.92–7.72)	<0.0001
**Serum-SDF1**	0.89 (0.51–1.42) ^#,^*	2.57 (1.69–3.23) ^♣^	4.77 (4.15–6.02) ^■^	7.04 (6.54–7.62)	<0.0001
**Urinary-SDF-1**	0.56 (0.28–0.92) ^#^	1.65 (1.32–1.93) ^♣^	3.52 (2.69–4.85) ^■^	5.27 (4.63–5.84)	<0.0001
**Serum-AOPP**	46.48 (39.35–48.22) ^#,^*	81.90 (69.46–87.86 ^♣^	145.02 (136.31–168.94) ^■^	294.48 (198.6–355.1)	<0.0001
**Urinary-AOPP**	32.63 (24.09–37.55) ^#,^*	52.00 (44.34–75.96) ^♣^	111.79 (86.78–132.06) ^■^	269.08 (174.2–328.6)	<0.0001
**Serum mtDNA**	14.98 (12.91–17.78) ^#,^*	9.84 (8.39–12.38) ^♣^	6.49 (5.28–7.12) ^■^	3.21 (1.85–4.52)	<0.0001
**Urinary mtDNA**	2.97 (1.70–4.66) ^ↂ,^*	4.00 (2.57–5.61) ^♣^	8.51 (7.06–10.10) ^■^	12.38 (10.78–13.99)	<0.0001
**IMT-CCA**	0.65 (0.62–0.71) ^#,^*	0.81 (0.77–0.92) ^♣^	0.97 (0.88–1.10) ^■^	1.22 (1.14–1.35)	<0.0001
**PI-ICA**	0.77 (0.69–0.89) ^#,^*	0.94 (0.79–1.02) ^♣^	1.04 (0.88–1.22) ^■^	1.29 (1.11–1.33)	<0.0001
**PI-MCA**	1(1–10.60 (0.56–0.67) ^#,^*	0.82 (0.68–0.89) ^♣^	0.92 (0.78–1.11) ^■^	1.17 (0.98–1.21)	<0.0001
**RI-ICA**	0.62 (0.55–0.65) ^#,^*	0.72 (0.66–0.78) ^♣^	0.89 (0.84–0.97) ^■^	1.23 (1.06–1.35)	<0.0001
**RI-MCA**	0.53 (0.51–0.55) ^#,^*	0.66 (0.61–0.70) ^♣^	0.96 (0.89–1.04) ^■^	1.21 (1.12–1.26)	<0.0001
**BHI**	1.07 (1.05–1.21) ^#,^*	0.79 (0.72–0.92) ^♣^	0.52 (0.49–0.62) ^■^	0.41 (0.37–0.48)	<0.0001

Clinical and biological data are presented as medians, and IQRs as for variables with skewed distribution. Significance between healthy controls and normoalbuminuric group, ^#^ *p* < 0.001; ^ↂ^ *p* = 0.04; significance between normoalbuminuric group and microalbuminuric group, ^⌂^ *p* = 0.008; ^♣^ *p* < 0.001; significance between microalbuminuric group and macroalbuminuric group, ^♦^ *p* = 0.005; ^■^ *p* < 0.001; ^⁑^ *p* = 0.001; significance between healthy controls vs. normoalbuminuric group vs. microalbuminuric group vs. macroalbuminuric group; ^*^ *p* < 0.001; eGFR: estimated glomerular filtration rate; UACR: urinary albumin/creatinine ratio; KIM-1/creat: urinary kidney injury molecule-1/creatinine ratio; NAG/creat: N-acetyl-β-(D)-glucosaminidase/creatinine ratio; Podocalyxin/creat: podocalyxin/creatinine ratio; Synaptopodin/creat: synaptopodin/creatinine ratio; SDF-1: stromal cell-derived factor-1; AOPP: advanced oxidation protein product; HbA1c: glycated hemoglobin; smtDNA: serum mitochondrial DNA; uDNA: urinary mitochondrial DNA; IMT-CCA: intima-media thickness common carotid artery; PI-ICA: pulsatility index internal carotid artery; PI-MCA: pulsatility index middle cerebral artery; RI-ICA: resistivity index internal carotid artery; RI-MCA: resistivity index middle cerebral artery; and BHI: breath-holding index. The cerebral ultrasound measurements are performed bilaterally. However, in the table, the values are only represented on the right side.

**Table 2 ijms-26-04481-t002:** Univariable regression analysis for mtDNA.

Parameter	Variable	R^2^	Coef β	*p*
**Serum mtDNA**	eRFG	0.437	0.258	<0.001
	UACR	0.223	−0.004	<0.001
	Serum P-selectin	0.573	−1.258	<0.001
	Serum SDF-1	0.564	−1.486	<0.001
	AOPP	0.459	−0.033	<0.001
	Synaptopodin	0.3207	−1.244	<0.001
	Podocalyxin	0.552	−1.219	<0.001
	KIM-1	0.446	−0.022	<0.001
	NAG	0.398	−0.418	<0.001
**Urinary mtDNA**	eRFG	0.227	−0.160	<0.001
	UACR	0.3105	0.004	<0.001
	Urinary P-selectin	0.5841	1.282	<0.001
	Urinary SDF-1	0.5261	0.029	<0.001
	Urinary AOPP	0.445	1.638	<0.001
	Synaptopodin	0.364	1.145	<0.001
	Podocalyxin	0.604	1.101	<0.001
	KIM-1	0.527	0.021	<0.001
	NAG	0.407	0.364	<0.001

UACR: urinary albumin/creatinine ratio; eGFR: estimated glomerular filtration rate; AOPP: advanced oxidation protein product; SDF-1: stromal cell-derived factor-1; KIM-1: urinary kidney injury molecule-1; and NAG: N-acetyl-β-(D)-glucosaminidase.

**Table 3 ijms-26-04481-t003:** Data provided by multivariable regression analysis for serum and urinary mtDNA.

Parameters	Variables	Coef β	*p*	95% CI	Prob > F	R^2^
**Serum mtDNA**	eGFR	0.1114	<0.0001	0.072 to 0.15	0.0000	0.656
	Serum P-selectin	−0.6282	<0.0001	−0.891 to −0.365		
	Serum SDF-1	−6.4502	<0.0001	−0.8 to −0.1402		
**Urinary mtDNA**	Podocalyxin	1.866	<0.0001	0.947 to 2.785	0.0000	0.628
	Urinary P-selectin	1.324	<0.0001	2.458 to 0.912	0.022	
	Urinary SDF-1	0.646	<0.0001	0.28 to 1.011	0.001	

eGFR: estimated glomerular filtration rate; SDF-1: stromal cell-derived factor-1.

**Table 4 ijms-26-04481-t004:** Univariable regression analysis for cerebral hemodynamic indices.

Parameter	Variable	R^2^	Coef β	*p*
**IMT-rCCA**	eGFR	0.41	−0.0116	<0.001
	UACR	0.36	0.0002	<0.001
	Serum P-selectin	0.62	0.06	<0.001
	Serum SDF-1	0.604	0.071	<0.001
	Serum AOPP	0.56	0.001	<0.001
	KIM-1	0.53	0.001	<0.001
	NAG	0.43	0.02	<0.001
	Synaptopodin	0.38	0.063	<0.001
	Podocalyxin	0.61	0.059	<0.001
**PI-rICA**	eGFR	0.27	−0.008	<0.001
	UACR	0.299	0.0002	<0.001
	Serum P-selectin	0.47	0.048	<0.001
	Serum SDF-1	0.47	0.057	<0.001
	Serum AOPP	0.41	0.001	<0.001
	KIM-1	0.38	0.008	<0.001
	NAG	0.341	0.016	<0.001
	Synaptopodin	0.299	0.05	<0.001
	Podocalyxin	0.46	0.046	<0.001
**PI-rMCA**	eGFR	0.38	−0.01	<0.001
	UACR	0.302	0.0002	<0.001
	Serum P-selectin	0.51	0.051	<0.001
	Serum SDF-1	0.52	0.622	<0.001
	Serum AOPP	0.45	0.0014	<0.001
	KIM-1	0.404	0.0009	<0.001
	NAG	0.402	0.018	<0.001
	Synaptopodin	0.32	0.054	<0.001
	Podocalyxin	0.49	0.05	<0.001
**RI-rICA**	eGFR	0.34	−0.011	<0.001
	UACR	0.53	0.0003	<0.001
	Serum P selectin	0.65	0.653	<0.001
	Serum SDF-1	0.67	0.794	<0.001
	Serum AOPP	0.61	0.0019	<0.001
	KIM-1	0.603	0.001	<0.001
	NAG	0.56	0.024	<0.001
	Synaptopodin	0.49	0.075	<0.001
	Podocalyxin	0.64	0.064	<0.001
**RI-rMCA**	eGFR	0.39	−0.013	<0.001
	UACR	0.37	0.0003	<0.001
	Serum P-selectin	0.79	0.079	<0.001
	Serum SDF-1	0.75	0.092	<0.001
	Serum AOPP	0.67	0.0022	<0.001
	KIM-1	0.67	0.001	<0.001
	NAG	0.62	0.028	<0.001
	Synaptopodin	0.49	0.082	<0.001
	Podocalyxin	0.77	0.767	<0.001
**BHI**	eGFR	0.55	0.015	<0.001
	UACR	0.25	−0.0002	<0.001
	Serum P-selectin	0.66	−0.071	<0.001
	Serum SDF-1	0.67	−0.086	<0.001
	Serum AOPP	0.56	−0.002	<0.001
	KIM-1	0.52	−0.071	<0.001
	NAG	0.55	−0.026	<0.001
	Synaptopodin	0.38	−0.072	<0.001
	Podocalyxin	0.63	−0.069	<0.001

UACR: urinary albumin/creatinine ratio; eGFR: estimated glomerular filtration rate; AOPP: advanced oxidation protein product; SDF-1: stromal cell-derived factor-1; KIM-1: urinary kidney injury molecule-1; NAG: N-acetyl-β-(D)-glucosaminidase; IMT-rCCA: intima-media thickness right common carotid artery; PI-rICA: pulsatility index right internal carotid artery; PI-rMCA: pulsatility index right middle cerebral artery; RI-rICA: resistivity index right internal carotid artery; RI-rMCA: resistivity index right middle cerebral artery; and BHI: breath-holding index.

**Table 5 ijms-26-04481-t005:** Multivariable regression analysis for the cerebral hemodynamic indices.

Parameter	Variable	Coef β	*p*	95% CI	Prob > F	R^2^
**IMT-rCCA**	eGFR	−0.0047	<0.0001	−0.0063 to −0.0031	0.000	0.702
	UACR	0.00007		0.00002 to 0.0001	0.002	
	Serum AOPP	0.0007		0.0002 to 0.001	0.002	
	Serum P-selectin	0.034		0.021 to 0.047	0.000	
	Synaptopodin	0.029		0.046 to 0.0122	0.001	
**PI-rICA**	eGFR	−0.002	<0.0001	−0.004 to −0.00007	0.042	0.511
	Serum P-selectin	0.023		0.0101 to 0.0366	0.001	
	Serum SDF-1	0.026		0.0096 to 0.0428	0.002	
**PI-rMCA**	eGFR	−0.004	<0.0001	−0.0061 to −0.0023	0.000	0.589
	Serum P-selectin	0.023		0.0106 to 0.036	0.000	
	Serum SDF-1	0.024		0.0085 to 0.0399	0.003	
**RI-rICA**	eGFR	−0.0018	<0.0001	−0.0035 to −0.00023	0.026	0.763
	UACR	0.00013		0.00008 to 0.00017	0.000	
	NAG	0.0058		0.002 to 0.0095	0.003	
	Serum P-selectin	0.0154		0.0033 to 0.028	0.013	
	Serum SDF-1	0.024		0.0098 to 0.387	0.001	
**RI-rMCA**	eGFR	−0.0017	<0.0001	−0.0032 to- 0.00031	0.018	0.848
	Synaptopodin	0.016		0.0274 to 0.0038	0.010	
	Podocalyxin	0.056		0.098 to 0.0153	0.007	
	KIM-1	0.0003		0.00008 to 0.00062	0.009	
	NAG	0.0044		0.00098 to 0.0078	0.013	
	Serum P-selectin	0.096		0.055 to 0.136	0.000	
	Serum SDF-1	0.27		0.0145 to 0.0394	0.000	
**BHI**	eGFR	0.007	<0.0001	0.0058 to 0.0091	0.000	0.789
	Podocalyxin	−0.061		−0.188 to −0.104	0.005	
	Serum P-selectin	−0.096		−0.142 to −0.0507	0.000	
	Serum SDF-1	−0.024		−0.038 to −0.0102	0.001	

UACR: urinary albumin/creatinine ratio; eGFR: estimated glomerular filtration rate; AOPP: advanced oxidation protein product; SDF-1: stromal cell-derived factor-1; KIM-1: urinary kidney injury molecule-1; NAG: N-acetyl-β-(D)-glucosaminidase; IMT-rCCA: intima-media thickness right common carotid artery; PI-rICA: pulsatility index right internal carotid artery; PI-rMCA: pulsatility index right middle cerebral artery; RI-rICA: resistivity index right internal carotid artery; RI-rICA: resistivity index right middle cerebral artery; and BHI: breath-holding index.

## Data Availability

The data that support the findings of this study are available from the corresponding author upon reasonable request.

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
