# Peer review of "Stromal Cell-Derived Factor-1, P-Selectin, and Advanced Oxidation Protein Products with Mitochondrial Dysfunction Concurrently Impact Cerebral Vessels in Patients with Normoalbuminuric Diabetic Kidney Disease and Type 2 Diabetes Mellitus"

_ijms, 2025, doi:10.3390/ijms26104481_

Round 1
Reviewer 1 Report
Comments and Suggestions for Authors
The aim of this study is to investigate a possible association between cerebral vascular remodeling and related functional decline, with biomarkers of inflammation, oxidative stress, and mitochondrial dysfunction in the early stages of chronic kidney disease in patients with type 2 diabetes mellitus. Although the findings could be interesting, the following main concerns should be taken into account:
In order to achieve a more accurate interpretation of the results and conclusions of the study, some demographic, anthropometric and clinical data may be relevant.
In particular:
1) Data on the age and gender distribution of patients and controls are important to interpret the associations;
2) The duration of diabetes is equally important.
3) Are the median serum creatinine values ​​exactly "1", without decimals, in all groups?
4) Since it is common for these patients to have elevated blood pressure values, do we have these data? (If yes, please include them in Table 1.) It is also important to know the proportion of hypertensive individuals in the sample.
5) Regarding carotid indices: are the median values ​​exactly 1? (It is difficult to interpret whether there are differences between groups based on these values.) These patients appear to be relatively "healthy" from the results of these tests. This should be mentioned in the text if appropriate.
6) What were the assumptions for the sample size calculation?
7) Is there any data that allows us to assess the cardiovascular risk of these patients? Certainly, patients with previous cerebrovascular disease were excluded, but were patients with previous coronary events included?
In summary: a better characterization of the patients and their controls with respect to demographic, anthropometric and clinical history variables will facilitate the interpretation by readers of the nature and importance of the associations found.
Author Response
Point-by-point reply
Reviewer 1
The aim of this study is to investigate a possible association between cerebral vascular remodeling and related functional decline, with biomarkers of inflammation, oxidative stress, and mitochondrial dysfunction in the early stages of chronic kidney disease in patients with type 2 diabetes mellitus. Although the findings could be interesting, the following main concerns should be taken into account:
In order to achieve a more accurate interpretation of the results and conclusions of the study, some demographic, anthropometric and clinical data may be relevant.
In particular:
- Data on the age and gender distribution of patients and controls are important to interpret the associations;
Thank you for your observation. The age and gender were added to the Cohort inclusion/exclusion criteria section and in Table 1.
- The duration of diabetes is equally important.
The duration of diabetes mellitus was also added in Table 1.
- Are the median serum creatinine values ​​exactly "1", without decimals, in all groups?
Thank you for your valuable comment.
There was an error in the statistical program which rounded the values to the closest whole number. The values were corrected in Table 1.
- Since it is common for these patients to have elevated blood pressure values, do we have these data?(If yes, please include them in Table 1.) It is also important to know the proportion of hypertensive individuals in the sample.
Thank you for your comment. The levels of blood pressure were added in Table 1. Cohort inclusion/exclusion criteria section: Hypertensive individuals (123) account for 66.84% of the patients included in the study.
5) Regarding carotid indices: are the median values ​​exactly 1? (It is difficult to interpret whether there are differences between groups based on these values.) These patients appear to be relatively "healthy" from the results of these tests. This should be mentioned in the text if appropriate.
Thank you for your valuable observation.
There was an error in the statistical program which rounded the values to the closest whole number. The values were corrected in Table 1.
- What were the assumptions for the sample size calculation?
Thank you for your pertinent observation. Sample size calculation was not performed as this is an exploratory study.
7) Is there any data that allows us to assess the cardiovascular risk of these patients? Certainly, patients with previous cerebrovascular disease were excluded, but were patients with previous coronary events included?
Thank you for your suggestion. Patients with coronary artery disease were excluded from the study.
Individuals exhibiting inadequate glycemic regulation (characterized by a HbA1c level exceeding 10%) or possessing a history or manifestations of cerebrovascular illness or coronary artery disease were excluded from the trial.
In summary: a better characterization of the patients and their controls with respect to demographic, anthropometric and clinical history variables will facilitate the interpretation by readers of the nature and importance of the associations found.

Reviewer 2 Report
Comments and Suggestions for Authors
This study investigated the interplay between inflammation, oxidative stress, mitochondrial dysfunction, and cerebrovascular remodeling in normoalbuminuric diabetic kidney disease (NADKD) in type 2 diabetes mellitus (T2DM) patients. The findings are novel and clinically relevant, but several aspects of the manuscript require improvement before it can be considered for publication. Below are detailed comments and suggestions for revision.
Major Concerns and Suggestions
1. While the study includes 184 patients, the subgroup sizes (e.g., 64 normoalbuminuric patients) are relatively small, which may limit the statistical power. Please discuss this limitation in the manuscript and clarify whether a sample size calculation was performed. The control group (39 healthy individuals) lacks detailed baseline characteristics, such as BMI, blood pressure, and lipid profiles. These factors could act as confounders. Please provide this information and discuss how potential confounding was addressed.
2. The manuscript reports multivariable regression models with high R² values, but it does not mention whether diagnostic checks (e.g., multicollinearity, residual analysis) were performed. Please include these details to strengthen the validity of the models. In addition, the analysis does not appear to account for key confounders such as medication use (e.g., ACE inhibitors, SGLT2 inhibitors), glycemic control, and lipid levels. Consider including these variables in the regression models or discussing their potential impact in the limitations section.
3. The tables are dense and difficult to interpret. Consider adding visual representations (e.g., box plots, scatter plots) to illustrate key findings, such as the differences in biomarkers across groups or their correlations with cerebrovascular indices.
Minor Concerns
The manuscript acknowledges that blood COâ‚‚ levels were not measured during the breath-holding maneuver, which may reduce the accuracy of the BHI. This limitation should be emphasized in the discussion.
Author Response
Point-by-point reply
Reviewer 2
This study investigated the interplay between inflammation, oxidative stress, mitochondrial dysfunction, and cerebrovascular remodeling in normoalbuminuric diabetic kidney disease (NADKD) in type 2 diabetes mellitus (T2DM) patients. The findings are novel and clinically relevant, but several aspects of the manuscript require improvement before it can be considered for publication. Below are detailed comments and suggestions for revision.
Major Concerns and Suggestions
- While the study includes 184 patients, the subgroup sizes (e.g., 64 normoalbuminuric patients) are relatively small, which may limit the statistical power. Please discuss this limitation in the manuscript and clarify whether a sample size calculation was performed.
Thank you for your pertinent observation. Sample size calculation was not performed as this is an exploratory study.
The control group (39 healthy individuals) lacks detailed baseline characteristics, such as BMI, blood pressure, and lipid profiles. These factors could act as confounders. Please provide this information and discuss how potential confounding was addressed.
The additional clinical and biological data were added in Table 1.
- The manuscript reports multivariable regression models with high R² values, but it does not mention whether diagnostic checks (e.g., multicollinearity, residual analysis) were performed. Please include these details to strengthen the validity of the models. In addition, the analysis does not appear to account for key confounders such as medication use (e.g., ACE inhibitors, SGLT2 inhibitors), glycemic control, and lipid levels. Consider including these variables in the regression models or discussing their potential impact in the limitations section.
Thank you for your valuable comment.
We used variance inflation factor (the vif command) after the regression to check for multicollinearity. Usually, a variable with a VIF value greater than 10 may warrant further research.
Limitation section-Third, the medication that the patients were under, such as ACEIs/ ARBs, statins, and SGLT2 inhibitors/GLP-1 agonists might have precluded an accurate interpretation of data.
- The tables are dense and difficult to interpret. Consider adding visual representations (e.g., box plots, scatter plots) to illustrate key findings, such as the differences in biomarkers across groups or their correlations with cerebrovascular indices.
Thank you for your pertinent observation. Results section: Correlation figures (1a-1f) between serum mtDNA and urinary mtDNA and serum and urinary SDF-1, P-selectin, and AOPPs, respectively, were added.
Minor Concerns
The manuscript acknowledges that blood COâ‚‚ levels were not measured during the breath-holding maneuver, which may reduce the accuracy of the BHI. This limitation should be emphasized in the discussion.
Thank you for your suggestion.
Discussion section-Our results concerning CVR and the decreased BHI, although limited by not measuring blood CO2 during the maneuver, show an indirect correlation with the increased levels of AOPPs, observation which supports the AOPP-induced endothelial dysfunction and impaired vasodilating capacity of the cerebral vessels.

Reviewer 3 Report
Comments and Suggestions for Authors
In the manuscript by Petrica et al., the authors potential association of cerebral vessels remodeling and its related functional impairment with biomarkers of inflammation, oxidative stress, and mitochondrial dysfunction in the early stages of DKD in type 2 diabetes mellitus (DM) patients. They evaluated various parameters like stromal cell-derived factor-1 (SDF-1), P-selectin, advanced oxidation protein products (AOPPs), urinary synaptopodin, podocalyxin, kidney injury molecule-1 (KIM-1), N-acetyl-β-(D)-glucosaminidase (NAG). It is an interesting associative study which cover many important information regarding potential SDF-1 and other markers with mitochondrial dysfunction and additional confirmation study using inhibitors. I have few comments for the authors to take into consideration.
- In a table, authors should add the additional details regarding ratio of male and females, smokers/non-smokers, any prior clinal history with other diseases for participated in the study.
- It would be interesting, if authors could add corelation graph should most important parameter findings of their study.
- Authors should also add the correlation graph of serum mtDNA vs urine mtDNA and other inflammatory markers like AOPP in serum vs urine and SDF-1 in serum vs urine. It important and relevant to known whether the kidney inframammary markers correlate with systemic inflammation.
Author Response
Point-by-point reply
Reviewer 3
In the manuscript by Petrica et al., the authors potential association of cerebral vessels remodeling and its related functional impairment with biomarkers of inflammation, oxidative stress, and mitochondrial dysfunction in the early stages of DKD in type 2 diabetes mellitus (DM) patients. They evaluated various parameters like stromal cell-derived factor-1 (SDF-1), P-selectin, advanced oxidation protein products (AOPPs), urinary synaptopodin, podocalyxin, kidney injury molecule-1 (KIM-1), N-acetyl-β-(D)-glucosaminidase (NAG). It is an interesting associative study which cover many important information regarding potential SDF-1 and other markers with mitochondrial dysfunction and additional confirmation study using inhibitors. I have few comments for the authors to take into consideration.
- In a table, authors should add the additional details regarding ratio of male and females, smokers/non-smokers, any prior clinical history with other diseases for participated in the study.
Thank you for your comment.
The clinical and biological data were added in the Cohort section and in Table 1.
- It would be interesting, if authors could add correlation graph should most important parameter findings of their study.
- Authors should also add the correlation graph of serum mtDNA vs urine mtDNA and other inflammatory markers like AOPP in serum vs urine and SDF-1 in serum vs urine. It important and relevant to known whether the kidney inflammatory markers correlate with systemic inflammation.
Thank you for your valuable observation.
Result section: Correlation figures (1a-1f) between serum mtDNA and urinary mtDNA and serum and urinary SDF-1, P-selectin, and AOPPs, respectively were added.

Round 2
Reviewer 2 Report
Comments and Suggestions for Authors
The authors have explained some of the potential biases
Author Response
Point-by-point reply
Authors have added the correlation graph but the equation r2 and p value is missing from the graph.
Thank you for your pertinent observation. The appropriate corrections have been made.
R2 and the P values have been added on Fig 1 (a-f).

Reviewer 3 Report
Comments and Suggestions for Authors
Authors have added the correlation graph but the equation r2 and p value is missing from the graph.
Author Response
Point-by-point reply
Reviewer 3
Authors have added the correlation graph but the equation r2 and p value is missing from the graph.
Thank you for your pertinent observation. The appropriate corrections have been made.
